# The Impact of Effective Communication on Perceptions of Patient Safety—A Prospective Study in Selected Polish Hospitals

**DOI:** 10.3390/ijerph19159174

**Published:** 2022-07-27

**Authors:** Janina Kulińska, Łukasz Rypicz, Katarzyna Zatońska

**Affiliations:** Department of Population Health, Faculty of Health Sciences, Wroclaw Medical University, 50-372 Wrocław, Poland; janina.kulinska@umw.edu.pl (J.K.); katarzyna.zatonska@umw.edu.pl (K.Z.)

**Keywords:** communication, patient safety, health care

## Abstract

Interpersonal communication plays a key role in the treatment process. It affects not only the patient’s satisfaction with the course of treatment, but also allows the achievement of the best outcome in the therapeutic process. The process of patient empowerment and putting them in the role of a partner in the therapeutic process makes it possible to build a relationship based on trust, kindness and empathy. The aim of the study is to show the relationship between a patient’s sense of safety and access to health information, friendly and empathetic treatment by medical staff and a relationship based on trust. The study is conducted on patients from public hospitals in Wrocław by using the author’s questionnaire. One of the five most important factors according to respondents is the sense of safety, as reported by more than half of the patients (54.4%)—this is the opinion of more than half of the patients (54.4%). The respondents assessed the quality of patient care as an average of M = 41.1/50 points. There is a strong positive correlation between the sense of security and the access to information (rho = 0.642), kind treatment (rho = 0.623), trust in medical staff (rho = 0.758) and satisfaction with hospital stay (rho = 0.758).

## 1. Introduction

Many years of transformations and reforms of the healthcare system, as well as mechanisms characteristic of the economy based on consumerism, have shaped the market of health services, the determinant of which is economic effectiveness. The main subject of the system is the patient, and meeting the patient’s expectations by providing a high-quality service is the main task of the provider. The quality of the service is judged largely based on the empathy of the staff, the kind treatment of the patients and their family, the respect for dignity and intimacy and a professional approach [1,2,3].

Patient empowerment builds a relationship based on trust and mutual respect between the patient and medical staff. This partnership model benefits both the clinician and the person being treated. The benefit for the clinician is that the patient is cooperative and well informed. The person being treated, on the other hand, has the opportunity to actively participate in the treatment based on comprehensive information—in terms of planning the entire treatment process and clarifying any doubts.

Communication in medical relations is based on two fundamental types of behavior. The first is connected with the instrumental aspect of curing the patient or reducing ailments and improving the quality of life. The second concerns the emotional and social aspects [4,5]. Many authors highlight the relationship between a patient’s sense of security and communication based on kindness, trust and empathy [6,7,8].

The quality of communication in the therapeutic process affects the way the patients experiences their illness and determines their attitudes toward it [9,10]. The most common reactions of persons affected by a disease are the emotions associated with loss of security. The perception of fear triggers the need for reassurance, which in turn reduces it and instils confidence. The patient subconsciously focuses on those statements made by staff that reduce their fears and provide a sense of security [11].

The relationship between trust and communication has been the subject of many studies [12,13,14,15,16]. The research shows that trust is the most valuable product of effective communication and a source of patient satisfaction—from the patient’s perspective, it can be considered in terms of trust in the competence, honesty and kindness of doctors [7]. The fundamental importance of communication in health care is demonstrated by the development of the new discipline as a distinct and recognized field of research in the United States. Two strands can be distinguished within it. The first focuses on the therapeutic role of communication. The second is related to health promotion, which belongs to the field of public health and is methodologically related to sociology and communication sciences [17].

Communication in health care is related to the interpersonal, organizational and media aspects. In this paper, based on a survey of hospitalized patients, the area of interpersonal communication and its relationship with patients’ perceived safety is discussed.

The aim of this study is to show the key role of communication between medical staff and the patient and access to information in the provision of high-quality health services, ensuring that the patient feels safe during hospitalization. The paper focuses on demonstrating the positive relationship between access to health information, kind and empathetic treatment of the patient by medical staff and a relationship based on trust and a sense of security.

## 2. Materials and Methods

The study was conducted between October 2017 and October 2018 using the author’s questionnaire. The survey included 778 patients hospitalized in 8 Wrocław public hospitals.

Data collection was based on a face-to-face interview with the patient. The interview was conducted by a member of the research team.

### 2.1. Questionnaire

The analysis concerned patients’ feelings during their hospital stay. The original questionnaire used in the study consisted of two parts: the socio-demographic questions and the essential part, the Patient Care Quality Assessment (PCQA) scale. Patient responses on the PCQA—10 scales were used to assess the quality of care in hospitals. Each statement (item) was rated on a Likert scale, 1—“definitely not”; 2—“rather not”; 3—“partially”; 4—“rather yes”, 5—“definitely yes”, with ratings reversed for the statement: “I feel that for the medical staff I am a disease entity/medical case”. The results of the item reliability analysis are presented in Table 1. The results of the internal consistency analysis of the patient care quality subscale indicate a high degree of consistency. Item total correlation coefficients ranged from 0.30 to 0.75. The item reliability coefficient after removing the item (alfa *) ranged from 0.856 to 0.900. These values indicate the high psychometric quality of the PCQA-10 scale.

### 2.2. Statistical Analysis

Statistical analysis was performed using the Statistica 13 (TIBCO Software Inc., Palo Alto, CA, USA). For all quantitative parameters (age, sum of scores of the scales used, etc.), the conformity of their distribution with the normal distribution was checked. Consistency was assessed using the Kolmogorov–Smirnov, Lilliefors and Shapiro–Wilk tests. The critical value, *p* < 0.05, was assumed to be the level of statistical significance. Mean values (M), standard deviation (SD), median (Me), lower (Q1) and upper (Q3) quartiles and extreme values, i.e., minimum (Min) and maximum (Max), were calculated for all quantitative parameters. For qualitative variables (nominal, e.g., gender and ordinal, e.g., education), counts (*n*) and percentages (%) were calculated and included in (multivariate) cross tabulations. Hypotheses of no correlation between two qualitative variables were verified using Pearson’s chi-squared test. A test result of *p* < 0.05 was taken as a significant correlation between variables. Parametric (Pearson’s r) or non-parametric (Spearman’s rho) correlations were used to assess relationships between variables.

### 2.3. Exclusion Criteria

The following groups of people were excluded from the study: patients whose hospitalization lasted less than 24 h, hospital staff, patients under the age of 18 years, friends and family of hospital staff, patients with whom effective communication could not be established or whose state of health did not allow them to participate in the study and patients who did not give their consent to participate in the study.

### 2.4. Ethical Considerations

The study was conducted in accordance with the tenets of the Declaration of Helsinki and guidelines of Good Clinical Practice (World Medical Association, 2013). The research project was approved by the independent Bioethics Committee at the Wroclaw Medical University (No KB-264/2016).

## 3. Results

The study included 445 women and 333 men, aged 18 to 90 years (mean M = 53.2; standard deviation SD = 17.2). The age distribution of the patients differed significantly from the normal distribution; it was left skewed (asymmetry coefficient A = −0.216). The majority of the studied patients were women (57.2%) who were younger than men by an average of 3 years (*p* < 0.01). Patients with secondary (37.3%) or higher (35.3%) education, aged 57 years and residents of Dolnośląskie Province (91.0%) were the majority, with more than 1/2 of the respondents living in Wrocław (50.4%). Most were economically active patients (46.4%) and pensioners (42.5%). The majority of the patients were treated in the hospital where the survey was conducted for the first time (57.2%) and their length of stay on the day of the survey was 2 to 3 days (31.0%) or a week or longer (29.6%).

To determine the element of the health service that is crucial during a hospital stay, respondents were asked to identify the top ten factors important to them during their hospitalization. At the top were factors related to the soft skills of medical staff. The kindness of the staff towards the patient was considered to be the most important aspect of the stay at the treatment facility (82.9% of indications). The availability of specialists, i.e., the option to ask questions and receive help and advice (75.4%), was also highlighted. One of the five most important factors according to respondents was the sense of safety—as reported by more than 1/2 of the patients (54.4%). Responses differed between the male and female groups. For women, staff kindness towards the patient (85.8% vs. 79.0%; *p* = 0.012) and the sense of security (59.8% vs. 47.1%; *p* < 0.001) were more important factors than for men.

On average, respondents rated the quality of patient care at M = 41.1/50. Standard deviation was SD = 6.4; Cronbach’s alpha α = 0.891 and the mean correlation of all PCQA scale items with the overall score was r = 0.464.

The components of patient care were rated by the hospitalized respondents on a five-point scale as follows: doctors treat me with respect and kindness (M ± SD: 4.4 ± 0.7); nurses treat me with respect and kindness (M ± SD: 4.4 ± 0.8); administrative staff treats me with respect and kindness (M ± SD: 4.2 ± 0.7); I get the impression that I am a “disease entity”/“medical case” for the hospital staff (M ± SD: 2.7 ± 1.3); the different stages of my treatment, what tests need to be done, what medicines to take and what their effects are were explained to me (M ± SD: 4.1 ± 1.1); I am kept informed about the progress of my treatment (M ± SD: 4.0 ± 1.0); the doctor is always available to me if needed (M ± SD: 4.0 ± 1.0); I feel that the staff is interested in my well-being (M ± SD: 3.9 ± 1.0); I feel safe in the hospital (M ± SD: 4.2 ± 0.8) and I trust the medical staff and believe that they want the best for me (M ± SD: 4.3 ± 0.8).

The sense of security was shown to correlate positively with access to information (rho = 0.642). This relationship is shown in Figure 1 and Figure 2. When assessing access to information, patients responded to the following statements: “The different stages of my treatment were explained to me”, “I am kept informed about the progress of my treatment” and “The doctor is always available to me if needed”. There is also a positive correlation between the sense of security and kind treatment (rho = 0.623) (Figure 3). The hypothesis that there is a relationship between the sense of security and trust in medical staff was also confirmed (rho = 0.758) (Figure 4). Furthermore, Figure 5 shows the relationship between safety and satisfaction with the hospital stay (rho = 0.758). There is also a positive correlation between the sense of security and not feeling that for the staff “I am a disease entity”/“medical case” (rho = 0.297), as well as with staff empathy (0.672) (Figure 6 and Figure 7).

The analysis of the results for the sense of security in different patient subgroups showed that feeling safe was more important for younger people and for women. It is less important for the unemployed.

## 4. Discussion

The interaction between the doctor and the patient has long-lasting effects related to regaining health or improving the quality of life. What is undeniable here is the impact of communication, which forms the basis for establishing a relationship based on partnership and trust. The most important effects of good relations include minimizing the patient’s anxiety and reducing the distress experienced, thereby helping the patient adopt appropriate attitudes towards the illness [9]. Jarosz notes that patients do not always verbalize their feelings of anxiety and loss of security. It is only by carefully analyzing their behavior, such as increasing aggression, withdrawal, excessive calmness and observation of other patients, that these non-verbal messages can be linked to increasing anxiety [11]. In the social-emotional aspect of the therapeutic process, the highly developed soft skills of the medical staff are necessary. A particularly desirable characteristic in reading a patient’s non-verbal communication will be the empathy of the staff.

In recent years, there has been a significant increase in the proportion of people who feel that patients are treated with kindness and care and who appreciate doctors’ commitment to their work. The health service component related to a qualitative relationship based on empathy, kindness and trust was also highlighted by the respondents of this study. The ten most important elements that affected patient’s stay in the hospital were, according to the respondents: kindness of staff towards the patient (82.9%), availability of specialists (75.4%), specialized equipment (59.8%), quick provision of health services (54.5%), the sense of security (54.4%), outstanding specialists (53.7%), general cleanliness (52.6%), hospital signposting and ease of moving around (40.1%), access to the facility (37.4%) and access to sanitary facilities in the room (37.3%). This means that almost half of the most important factors relate precisely to the human element. Patients also confirmed this in a question aimed at determining the strongest and weakest sides of the hospital in which they were hospitalized. Respondents unanimously agreed that the greatest asset of the studied facilities was the staff they employed.

The key role of the medical staff was also indicated in Batko’s study conducted on a group of 600 patients using health services in the Śląskie Province, in which he reported that among the most important factors affecting image building, the patients mentioned were a comprehensive medical offer, competent and experienced staff and care for the patient [18].

In Poland, the medical profession is still respected because of the social functions it fulfils—it provides a sense of security and brings aid to the population [19]. Expectations towards representatives of this professional group therefore concern not only professional knowledge and skills, but also ethical behavior consistent with the medical oath [19,20]. In a study conducted by Cook et al., it was shown that trust in the provider and satisfaction with their services are linked [21]. Our own research allowed the author to accept the hypothesis that there is a positive relationship between the patient’s sense of security and kindness and empathy of the medical staff and the degree of trust.

Krot talks about the multidimensionality of trust, saying that it is a construct consisting of dimensions relating to competence, honesty and kindness. Krot presents results that indicate that respondents are most convinced about the kindness of doctors (mean 3.53); a slightly lower level of trust is declared in relation to their competence, honesty and reliability (mean 3.41) [22]. Trust in the doctor–patient relationship is the basis of an effective relationship [23,24], allowing the achievement of the intended therapeutic effect. European Trusted Brands studies have shown a relatively low level, compared to the average, of trust Poles have in the medical profession. Europeans trust nurses (82%) and pharmacists (80%), while doctors are trusted by 76% of Europeans. Poles, in turn, place pharmacists in the first place in terms of trust among medical professions (78%), followed by nurses (76%), and only after them doctors (57%) [25]. The results obtained in the Wrocław hospitals are more optimistic. Patients unanimously stated that they trusted medical staff and believed that they wanted the best for them. They rated this trust on a five-point scale at 4.3 (M ± SD, 4.3 ± 0.8). Certainly, patients’ assessments were influenced by the circumstances in which they were made. A hospitalized patient who is cared for and receives help in their illness does not make a hypothetical judgement, as is the case in general trust reports, but is in a real situation where they have contact with a specific doctor. This is also supported by a study conducted by Kozimala et al. using the Anderson and Dedrick scale among 597 respondents participating in the survey. Patients assessed trust in the doctor and compliance with the recommendations at 3.7873–75.21% of patients trusted the doctor about the treatment method [26]. Additionally, the study by Krajewska et al., using the same scale, showed that doctors were trusted and their advice was followed by 91.7% of Polish patients [27,28].

The relationship between the doctor and the patient depends both on the doctor’s professional competence, knowledge, experience and communication skills, but also on the conditions in which they admit the patient, the team assisting the doctor, the available equipment, medications and the opinions of other patients [19] (pp. 126–127). The BioStat survey, “Perception of doctors and their problems in their professional work”, conducted in 2018 on a group of 1000 people confirmed that the majority of Poles (64%) support a positive opinion about the therapeutic competence of doctors [29]. In the author’s own research, patients also most frequently mentioned competent, professional staff as one of the hospital’s greatest advantages.

Differences in the opinions of respondents in the BioStat report, on the other hand, appear in the assessment of doctors’ soft skills—medical competencies are definitely rated higher than communication skills. In assessing doctors’ commitment to their work, 43% of respondents believed that doctors were committed to their work and 45% believed the opposite [29]. Doctors have the ability to listen according to only 46% of respondents, and the skill of communicating relevant information about the patient’s health according to 51%. Lesiak’s research using participant observation analyzed the doctor–patient relationship, which showed that this professional group is often characterized by a lack of language skills in conversations with patients. The use of the so-called baby talk with older patients only makes it harder for them to understand the message by making it sound childish [29]. Barczykowska-Tchórzewska et al. stressed that was is not enough for the patient to receive an instructive message on how to proceed, but they also expected emotional qualities from the relationship: expression, reassurance, understanding and acceptance [30]. The author’s own research confirmed that patients required more interest and involvement on the part of medical staff. On a five-point scale, at 3.9 (M ± SD, 3.9 ± 0.8), patients rated the response “I feel that the staff are interested in my well-being”, but other questions related to empathy, kindness and showing respect to the patient scored above the mean of 4.0 (“I am treated with respect and kindness by the doctors”—M ± SD: 4.4 ± 0.7; “the different stages of my treatment were explained to me”—M ± SD: 4.1 ± 1.1; “I am kept informed about the progress of my treatment”—M ± SD: 4.0 ± 1.0). According to the patients in Wrocław, the relationship with the medical staff was therefore satisfactory. The patients emphasized that they were aware that the quality of staff’s work was affected by the systemic constraints that health care as a whole faces. One obstacle to a satisfactory doctor–patient relationship is the shortage of medical staff in the country—according to the Organisation for Economic Cooperation and Development report, there are 25 employees per 1000 inhabitants and only 2.3 doctors [31]. This creates, among other things, the problem of long waiting times for appointments that are also too short. A study commissioned by LekSeek Polska showed that as much as sixteen minutes out of a twenty-minute medical visit is taken up by organizational activities. It is unrealistic to expect the remaining four minutes to constitute a satisfactory appointment [32]. Other problems that patients have shown to be aware of both in the above-mentioned report and in the research presented in this paper are excessive bureaucracy, a lot of “paperwork”, too many patients per doctor and working in several institutions at the same time.

According to the LekSeek survey, as many as 41% of those surveyed believed that doctors did not have time for further training to improve their competencies. Additionally, the quantitative study on the sample of 1608 young doctors conducted by the Chamber of Physicians and Dentists in connection with the project entitled “Nationwide training in the functioning of the health care system and in communication, cooperation and relationship building skills for doctors starting their career” showed that the vast majority of respondents (70%) had never participated in training in soft skills, while almost all the respondents (97%) stated that soft skills are at least as important as professional knowledge and skills (13% were of the opinion that they are even more important) [33].

## 5. Conclusions

The following conclusions can be drawn from the results obtained. In order to satisfy the modern patient, it is necessary to provide them with a quality health service based on a medical aspect, which focuses on the improvement of health, and a social-emotional aspect, which directly influences the satisfaction with the service received. A relationship based on partnership, trust and effective communication between the patient and the medical staff positively influences the patient’s perception of the illness and strengthens their sense of security. It should be emphasized that patients perceived some systemic constraints that adversely affected the relationship between the patient and staff—a point that was repeatedly stressed during the survey. The development of soft skills of the medical staff is an important pillar of a medical facility’s functioning; therefore, the managers of those entities should give priority to the implementation of an effective, or improvement of an already existing, motivational program oriented towards building social competencies of the employees.

## Figures and Tables

**Figure 1 ijerph-19-09174-f001:**
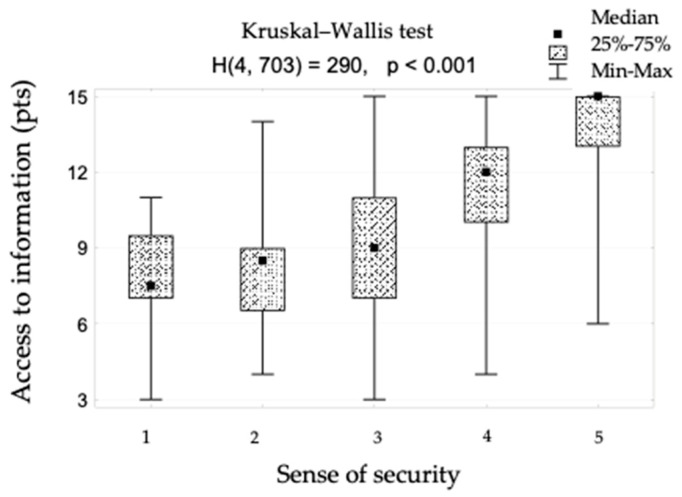
Access to information in patient groups differing in the sense of security (1—definitely not, 2—rather not, 3—partially, 4—rather yes, 5—definitely yes) and the result of the significance test (“H” indicates the number of degrees of freedom.).

**Figure 2 ijerph-19-09174-f002:**
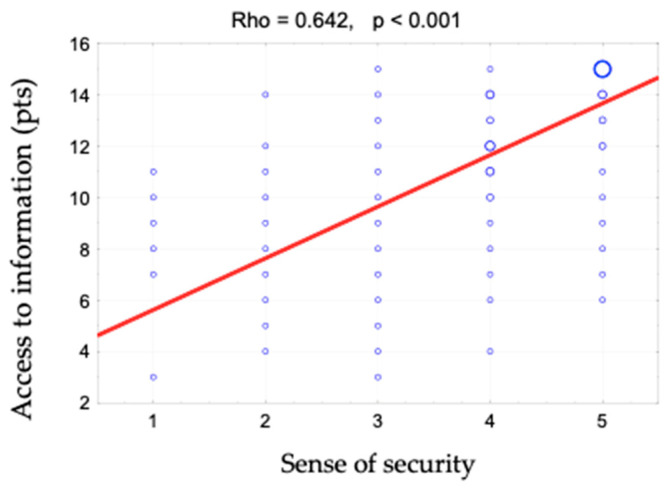
Correlation diagram between the sense of security (1—definitely not, 2—rather not, 3—partially, 4—rather yes, 5—definitely yes) and access to information and the value of Spearman’s rank correlation coefficient (rho).

**Figure 3 ijerph-19-09174-f003:**
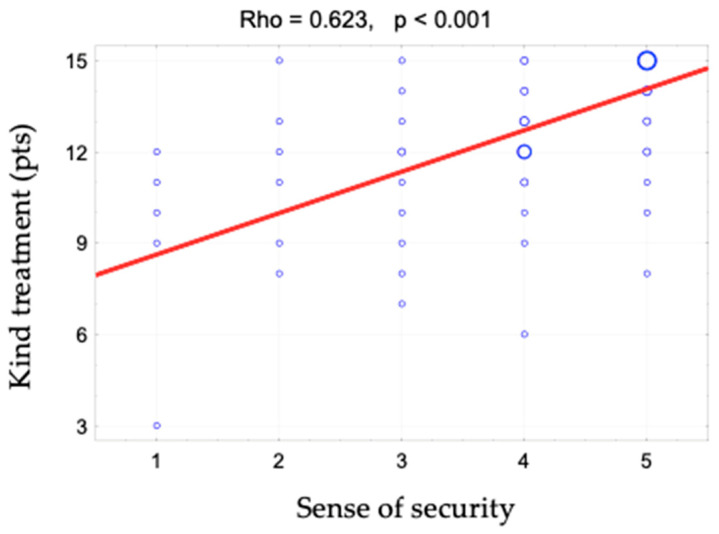
Correlation diagram between the sense of security (1—definitely not, 2—rather not, 3—partially, 4—rather yes, 5—definitely yes) and kindness and the value of Spearman’s rank correlation coefficient (rho).

**Figure 4 ijerph-19-09174-f004:**
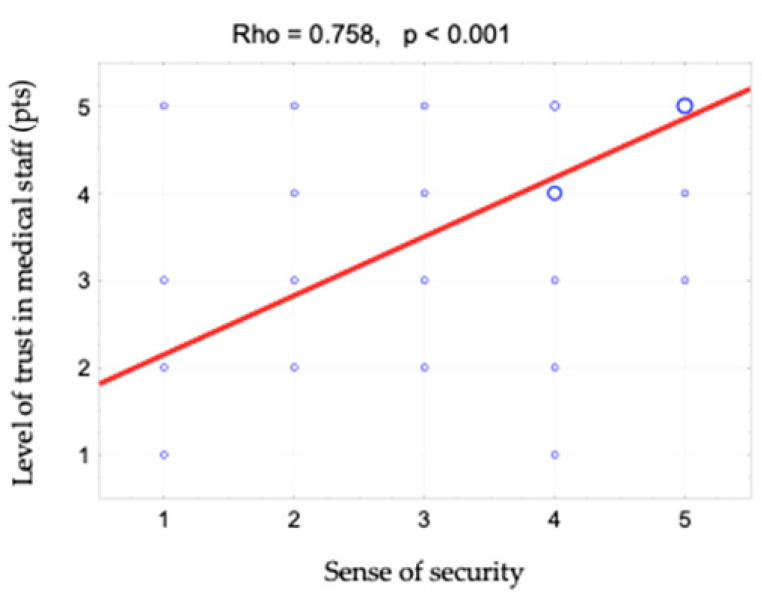
Correlation diagram between the sense of security (1—definitely not, 2—rather not, 3—partially, 4—rather yes, 5—definitely yes) and level of trust in medical staff and the value of Spearman’s rank correlation coefficient (rho).

**Figure 5 ijerph-19-09174-f005:**
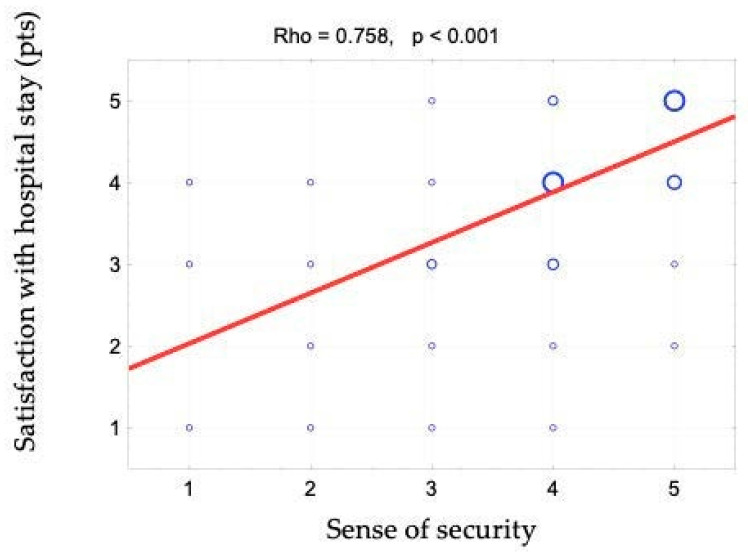
Correlation diagram between the sense of security (1—definitely not, 2—rather not, 3—partially, 4—rather yes, 5—definitely yes) and satisfaction with hospital stay and the value of Spearman’s rank correlation coefficient (rho).

**Figure 6 ijerph-19-09174-f006:**
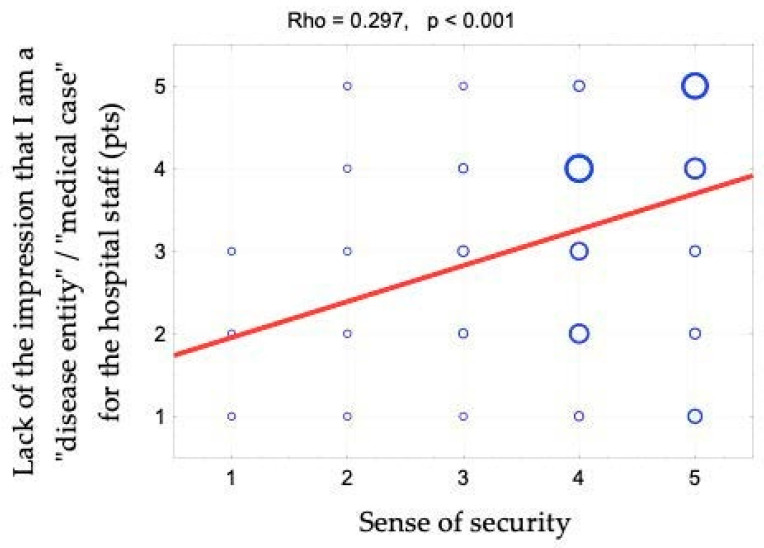
Correlation diagram between the sense of security (1—definitely not, 2—rather not, 3—partially, 4—rather yes, 5—definitely yes) and the feeling of a lack of the impression that I am a “disease entity”/“medial case” for the hospital staff and the value of Spearman’s rank correlation coefficient (rho).

**Figure 7 ijerph-19-09174-f007:**
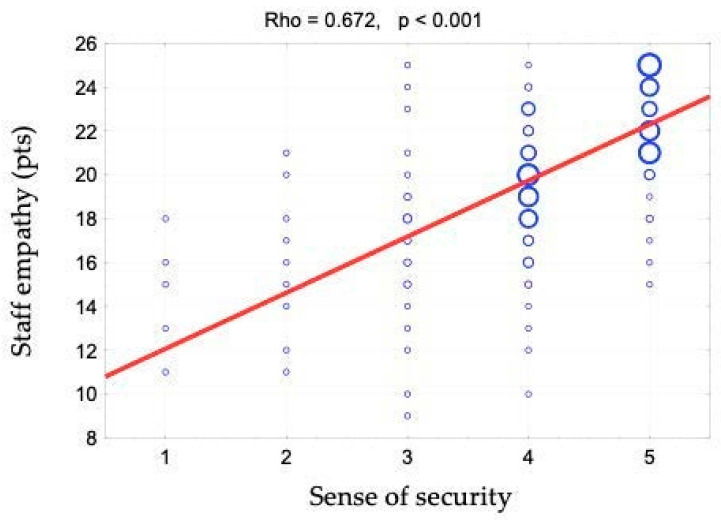
Correlation diagram between the sense of security (1—definitely not, 2—rather not, 3—partially, 4—rather yes, 5—definitely yes) and staff empathy and the value of Spearman’s rank correlation coefficient (rho).

**Table 1 ijerph-19-09174-t001:** Analysis of the item reliability of the Patient Care Quality Assessment (PCQA) scale.

Item	Me *	SD *	R *^,^**	α *
A. Doctors treat me with respect and kindness	36.6	6.0	0.618	0.867
B. Nurses treat me with respect and kindness	36.7	5.9	0.600	0.867
C. Administrative staff treats me with respect and kindness	36.9	6.0	0.522	0.872
D. I get the impression that I am a “disease entity”/“medical case” for the hospital staff *	37.7	5.9	0.295	0.900
E. The different stages of my treatment were explained to me	36.9	5.7	0.616	0.866
F. I am informed about the progress of my treatment	37.0	5.7	0.678	0.860
G. The doctor is always available to me if needed	37.1	5.7	0.718	0.857
H. I feel that the staff are interested in my well-being	37.2	5.7	0.730	0.856
I. I feel safe in the hospital	36.9	5.8	0.754	0.856
J. I trust the medical staff and believe that they want the best for me	36.7	5.8	0.741	0.858

* Me—median, SD—standard deviation, r—correlation coefficient, α—reliability coefficient alpha. ** Ranges of absolute values of the r-correlation coefficient: <0.2: weak correlation; 0.3–0.5: sufficient correlation; 0.6–0.7: moderate correlation; 0.8–0.9: very strong correlation; 1: perfect correlation.

## Data Availability

Not applicable.

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
