# Peer review of "The Impact of Effective Communication on Perceptions of Patient Safety—A Prospective Study in Selected Polish Hospitals"

_ijerph, 2022, doi:10.3390/ijerph19159174_

Round 1
Reviewer 1 Report
The importance of the doctor-patient relationship cannot be overemphasized. Through systematic research such as this study, it is possible to prove the basic knowledge that a doctor should have scientifically. This paper has no methodological problems, and the results and discussion are well done. Here are some suggestions for figures and tables:
1. Table 1: r is the correlation coefficient, and it would be good to clearly describe in the table what each item and correlation was seen.
2. Figure 1: What does H(4,703)=290 mean in the title?
3. It would be better if the Conclusion is written in lines rather than in a list format.
Author Response
Dear Reviewer,
thank you for your valuable comments. I hope that the proofreading will make the work acceptable.
On behalf of the authors, thank you.

Reviewer 2 Report
This original article addresses a subject of great interest: the degree of safety perceived by hospitalised patients. It deserves to be published in IJERPH, however, there are several points in which improvements should be made to strengthen the manuscript.
Please make adjustments on the following items.
Title
The study assessed the perception of safety, not safety itself. This difference should also be visible in the title.
Abstract
Line 9 – I recommend an impersonal statement and the avoidance of the pronoun “you”.
Lines 12 – 14 – Rephrase the aim of the study to increase clarity and highlight the subject, namely the sense of security, by mentioning it first. The current form confuses the reader due to the list of studied items in relation to the sense of safety.
Lines 15-17 –The sentence “Questions regarding the quality of patient care.” does not have a predicate. In addition, the next sentence needs rephrasing. It mentions “five most important factors are” and only states one factor “the sense of security”. Actually, as mentioned in Discussion, the sense of security is the fifth most important element that affects patient's stay in the hospital.
Introduction
Line 29 – The use of the pronoun “their” implies a plural, use “patients and their …”.
Lines 32 – 35 – The phrase is too long and confusing, mentioning first the clinician, then the patient and again the clinician. Please use shorter sentences to increase clarity.
Line 41 – Use the plural “the patients experience their illness”
Lines 44 – 45 – Rephrase “A consequence of the fear is the need for reassurance to inspire confidence and reduce anxiety” to increase clarity.
Materials and Methods
A more thorough description of the questionnaire would be helpful. What was the structure of the questionnaire? Did the questionnaire contain any other items than the ones mentioned in Table 1 A-J?
There is no statement about the answer collection procedure. Was there an interview (if so, who conducted it?) or the patients received a paper form?
Line 74 – Mention the meaning of the abbreviation OJOP and add this method to references.
Lines 74 –95 - Mention references for each statistic test or scale.
Table 1 – M was introduced in line 88 as the abbreviation for mean values and now the same abbreviation is used for median (line 112). Please modify it. Please use the singular for “coefficient alpha”.
Results
Line 115 – The minimum age of the participants (15 years) is under the limit set in Materials and methods. In the exclusion criteria you mentioned “patients under the age of 18”. Please verify and make the correction.
Lines 130 – 132 – Increase the clarity of this statement that also appears in the Abstract ”The top five …..(54.4%).”
Lines 136 – 138 – The data mentioned in this paragraph is not visible in Table 1.
Figures
Please adjust the figures and legends to have all the mandatory information on each of them (e.g. Measurement units are present in Figure 1, but are absent in Figures 2-7. The word “yes” is missing from the 4th item on the OX axis in Figure 1. The elements on the OX axis have names in Figure 1 and the same elements have numbers in Figure 2, etc.)
Ensure the proper correlation between the title of Figure 6 and the OY axis label.
Discussion
In the text there is no mentioning of reference number 19.
Lines 283 – 285 – In what manner was collected the information regarding that “patients were aware that the quality of staff's work was affected by the systemic constraints that health care as a whole faces”? How was it quantified? Materials and Methods lack information.
Line 287 – Use the complete name of the institution instead of the abbreviation OECD.
Conclusions
The results in current presentation form do not directly support conclusion number 3. Please revise.
References
Please style citations and references according to IJERPH.
Author Response

(The authors gave the same response as above.)

Round 2
Reviewer 2 Report
Dear Authors,
I appreciate the attention you have shown in addressing my comments.